# Assessing fetal growth in Africa: Application of the international WHO and INTERGROWTH-21st standards in a Beninese pregnancy cohort

Emmanuel Yovo[1]*, Manfred Accrombessi[1,2], Gino Agbota[1,3], Alice Hocquette[4], William Atade[1], Olaiitan T. Ladikpo[1], Murielle Mehoba[1], Auguste Degbe[1], Ghyslain Mombo-Ngoma[5,6,7,8], Achille Massougbodji[1], Nikki Jackson[9], Nadine Fievet[10], Barbara Heude[11], Jennifer Zeitlin[4], Valérie Briand[8,12]*

1 Institut de Recherche Clinique du Bénin (IRCB), Abomey-Calavi, Benin, 2 Disease Control Department, Faculty of Infectious and Tropical Diseases, London School of Hygiene and Tropical Medicine, London, United Kingdom, 3 IRD UMI 233 TransVIHMI- UM-INSERM U1175, Montpellier, France, 4 Université de Paris, CRESS, Obstetrical Perinatal and Pediatric Epidemiology Research Team, EPOPé, INSERM, INRA, Paris, France, 5 Centre de Recherches Médicales de Lambaréné (CERMEL), Lambaréné, Gabon, 6 Institute of Tropical Medicine, University of Tübingen, Tübingen, Germany, 7 Department of Tropical Medicine, Bernhard Nocht Institute for Tropical Medicine, University Medical Center Hamburg-Eppendorf, Hamburg, Germany, 8 I. Department of Medicine, University Medical Center Hamburg-Eppendorf, Hamburg, Germany, 9 Department of Obstetrics and Gynaecology, Oxford University, Oxford, United Kingdom, 10 Université de Paris, MERIT, IRD, Paris, France, 11 INSERM, UMR 1153, Centre for Research in Epidemiology and StatisticS (CRESS), "EArly life Research on later Health" (EARoH) team, Paris, France, 12 IRD, Inserm, Université de Bordeaux, IDLIC team, UMR 1219, Bordeaux, France

* emkoffiyovo@gmail.com (EY); valerie.briand@ird.fr, valerie.briand@gmail.com (VB)

**Data Availability Statement:** All relevant data are within the paper and its Supporting Information file.

## Abstract

### Background

Fetal growth restriction is a major complication of pregnancy and is associated with stillbirth, infant death and child morbidity. Ultrasound monitoring of pregnancy is becoming more common in Africa for fetal growth monitoring in clinical care and research, but many countries have no national growth charts. We evaluated the new international fetal growth standards from INTERGROWTH-21st and WHO in a cohort from southern Benin.

### Methods

Repeated ultrasound and clinical data were collected in women from the preconceptional RECIPAL cohort (241 women with singleton pregnancies, 964 ultrasounds). We modelled fetal biometric parameters including abdominal circumference (AC) and estimated fetal weight (EFW) and compared centiles to INTERGROWTH-21st and WHO standards, using the Bland and Altman method to assess agreement. For EFW, we used INTERGROWTH-21st standards based on their EFW formula (IG21st) as well as a recent update using Hadlock's EFW formula (IG21hl). Proportions of fetuses with measurements under the 10th percentile were compared.

**Funding:** This work was supported by the French Agence Nationale de la Recherche (grant number ANR-13-JSV1-0004) and the Fondation Simone Beer under the auspices of the Fondation de France (grant number 00074147), the funding is granted to VB. EY received Idex Travel Scholarship, Excellence Initiative from Bordeaux University in France for MPH studies. The funders had no role in study design, data collection and analysis, decision to publish, or preparation of the manuscript.

**Competing interests:** The authors have declared that no competing interests exist.

**Abbreviations:** FGR, Fetal growth restriction; SGA, Small-for-gestational age; GA, Gestational Age; LMICs, low- and middle-income countries; HIV, Human immunodeficiency virus; HICs, high-income countries; INTERGROWTH-21st, The International Fetal and Newborn Growth Consortium for the 21st Century; WHO, World Health Organization; US, ultrasound scan; RECIPAL, retard de croissance intra-utérin et paludismse; LMP, last menstrual period; wg, weeks of gestation; CRL, crown-rump length; EFW, estimated fetal weight; HC, head circumference; AC, abdominal circumference; FL, femur length; SD, standard deviation; ANC, antenatal care.

## Results

Maternal malaria and anaemia prevalence was 43% and 69% respectively and 11% of women were primigravid. Overall, the centiles in the RECIPAL cohort were higher than that of INTERGROWTH-21st and closer to that of WHO. Consequently, the proportion of fetuses under 10th percentile thresholds was systematically lower when applying IG21st compared to WHO standards. At 27–31 weeks and 33–38 weeks, respectively, 7.4% and 5.6% of fetuses had EFW <10th percentile using IG21hl standards versus 10.7% and 11.6% using WHO standards.

## Conclusion

Despite high anemia and malaria prevalence in the cohort, IG21st and WHO standards did not identify higher than expected proportions of fetuses under the 10th percentiles of ultrasound parameters or EFW. The proportions of fetuses under the 10th percentile threshold for IG21st charts were particularly low, raising questions about its use to identify growth-restricted fetuses in Africa.

## Introduction

Fetal growth restriction (FGR), or poor growth of a fetus during pregnancy, is associated with an increased risk of infant mortality and morbidity at birth and in childhood [1–3]. Therefore, early detection and surveillance of growth-restricted fetuses can contribute to reducing the short- and long-term consequences of FGR [4, 5]. Because defining and identifying FGR is difficult, small-for-gestational age (SGA) is commonly used as a proxy. SGA is defined as a fetal weight or birthweight below the 10th centile for a given gestational age (GA) according to a reference chart [6, 7]. In low- and middle-income countries (LMICs), about 20% of newborns are considered to be SGA at birth and account for 25% of neonatal deaths [3]. Maternal undernutrition, gestational hypertension, and infectious diseases (e.g. malaria, HIV) are among the main causes of FGR and SGA in these countries [8–10].

In Africa, with limited exceptions [11], the reference charts used for fetal growth monitoring come from high-income countries (HICs), where populations have different characteristics and risk factors for FGR [8, 12]. Recently, two international fetal growth standards were developed for global use: the INTERGROWTH-21st and the World Health Organization (WHO) [13, 14] standards. Their comparison to national reference charts worldwide has yielded contradictory results [15–21]. However, few studies have been carried out in African populations [22]; assessment of these charts requires accurate GA estimates by ultrasound scans (US) along with serial fetal biometric parameters, which are still uncommon in the African context.

The RECIPAL study, which established a preconceptional prospective cohort of pregnant women in Benin, offers an opportunity to contribute to the assessment of these new standards in Africa. Our objective was to compare fetal growth in the RECIPAL cohort based on models of fetal biometric measurements and estimated fetal weight (EFW) centiles with the WHO and INTERGROWTH-21st standards as well as the one existing African EFW chart from Tanzania [11]. Given the high prevalence of malaria, anemia and under-nutrition among pregnant women in Benin, we hypothesized that the proportion of fetuses classified as SGA by each

international prescriptive chart would be greater than 10%, as these charts were developed in low-risk pregnant women.

## Methods

### Study design, population, and procedures

The RECIPAL study was conducted in Sô-Ava and Abomey-Calavi districts, south Benin, in 2014–2017. Briefly, women of reproductive age (18–45 years old) were recruited at the community level and followed monthly for a maximum period of 24 months until becoming pregnant [23]. During the monthly home visit after enrollment, the first day of last menstrual period (LMP) was recorded and a urinary pregnancy test was performed. The subsample of women who became pregnant was then followed monthly from early pregnancy to delivery. Data on risk factors for FGR such as malaria, HIV, gestational hypertension, malnutrition, anaemia, alcohol consumption, smoking, and urogenital infection were collected either at recruitment before conception or monthly during pregnancy. During pregnancy, women received intermittent preventive treatment with sulfadoxine-pyrimethamine and an insecticide-treated net, plus folic acid and iron supplementation. In case of malaria, women were treated with quinine (in the 1st trimester) or artemisinin-based combinations (in the 2nd and 3rd trimesters). Newborns were weighed within 1 hour of birth using an electronic digital scale with an accuracy of 2g (SECA 757; SECA, Germany).

The RECIPAL study received ethical approval from the Beninese Ethics Committee of the Institut des Sciences Biomédicales Appliquées and Ministry of Health. All participants gave informed written consent before enrollment in the cohort.

### Ultrasound examination

The first US for dating the pregnancy was performed between 9 and 13 weeks of gestation (wg) (±1week); dating was based on the crown-rump length (CRL) measurement using the Robinson's formula [24]. GA was based on the LMP if the difference between the LMP and CRL was less than 7 days or on CRL if the difference was >7. Then, four additional standardized USs were performed every 6 weeks (±1week) for fetal growth monitoring, so that the possible ranges of GA were 15–20, 21–26, 27–32 and 33–38wg. At each US, head circumference (HC), abdominal circumference (AC), and femur length (FL) were measured twice in two separate subsequent images. Fetal weight was estimated based on HC, FL, and AC parameters using both the Hadlock formula [25] and the INTERGROWTH-21st formula [26]. USs were performed by four skilled obstetrician-gynaecologists using a portable ultrasound system (high-resolution ultrasound system, 5–2 MHz C60 abdominal probe; Sonosite M-TURBO, Washington State, USA). Throughout the study, 10% of the images were reviewed by a senior obstetrical sonographer to verify that the measurements fulfilled the INTERGROWTH-21st guidelines [27].

### Statistical analysis

For each US and each set of fetal measurements, Bland and Altman plots were used to assess the intra-operator variability. After selecting measurements that fell within acceptable ranges for each parameter [28], the mean was calculated and used for comparison with the reference values. The few sets of measurements that fell outside the acceptable ranges were mainly due to data entry errors and were corrected by returning to the source data, then included in data analysis.

Centiles for AC, HC, FL, and EFW were derived from 15 to 38 wg with the RECIPAL data using quantile regression analysis, following the WHO modelling approach [13]. In our study, RECIPAL EFW were estimated with both the Hadlock formula [25] and the INTER-GROWTH-21st formula [14] (see below). The quantile regression calculates quantiles (ie percentiles) directly from the observed measurements without making assumptions about their distribution. To assess the validity of the regression model applied to the RECIPAL data, the proportion of fetuses with observed values below the threshold of each percentile (i.e., 3th, 5th, 10th, 25th, 50th, 75th, 90th, 95th, and 97th) was calculated.

The 10th, 50th, and 90th centiles for AC, HC, FL, and EFW centiles from RECIPAL cohort were compared with both INTERGROWTH-21st and WHO centiles, as well as those from a recent EFW growth chart that was developed in Tanzania [11]. Two INTER-GROWTH-21st standards for EFW were used: the original ones using the INTER-GROWTH-21st formula for calculating EFW (hereafter denoted IG21st) [26], and recently published standards using the Hadlock formula for calculating EFW (hereafter denoted IG21hl) [29]. This new standard follows research showing Hadlock formula to be more accurate for the prediction of fetal weight. Therefore, EFW were estimated with the Hadlock formula for comparison between RECIPAL, IG21hl, WHO and the Tanzanian standards, or with the INTERGROWTH-21st formula for comparison between RECIPAL and IG21st. The agreement was assessed using the Bland and Altman method of differences analysis of two quantitative measurements [30, 31]. For instance, the percentage difference of the 10th centile of AC compared between RECIPAL and WHO was calculated as follows: $[(AC10th\_WHO - AC10th\_RECIPAL)/Mean\ AC10th] * 100$ where $AC_{10th}\_WHO$ is the value of the 10th centile for AC based on WHO standards, $AC_{10th}\_RECIPAL$ is the value of the 10th centile for AC based on RECIPAL charts, and Mean $AC_{10th}$ is the mean of $AC_{10th}$ between WHO and RECIPAL. A negative percentage difference means that the RECIPAL centile is higher than that of WHO. Percentage differences were plotted by GA and the bias (mean percentage differences between the paired data) was calculated. The closer these differences are to zero, the more similar the paired data are to each other.

The proportion of fetuses with HC, AC, FL, and EFW less than the 10th centile of INTER-GROWTH-21st and WHO standards, as well as EFW for the Tanzania standards at 27–32wg and at 33–38wg was calculated.

Stata version 13 for Windows (Stata Corp., College Station, TX) was used for all statistical analyses.

## Results

The RECIPAL study included 411 pregnant women, of whom 273 (66.4%) were followed until delivery and 254 had at least one US between 27–38 wg and gave birth to a live singleton baby (Fig 1). These women were included in the calculation of the proportion of fetuses with AC, HC, FL and EFW centiles below the 10th. Fetal growth modeling was carried out on the 241 (92%) women who underwent the four scheduled growth monitoring USs.

The first US for dating the pregnancy was performed at a mean of 11 wg. The 241 women received the four scheduled USs within the expected GA windows, at 16, 22, 28, and 34 wg respectively (S1 Fig). Overall, 964 ultrasounds were performed for fetal growth monitoring.

### Women's demographic characteristics and main risk factors for FGR

The mean maternal age was 26.7 years; 11% of women were primigravid (Table 1).

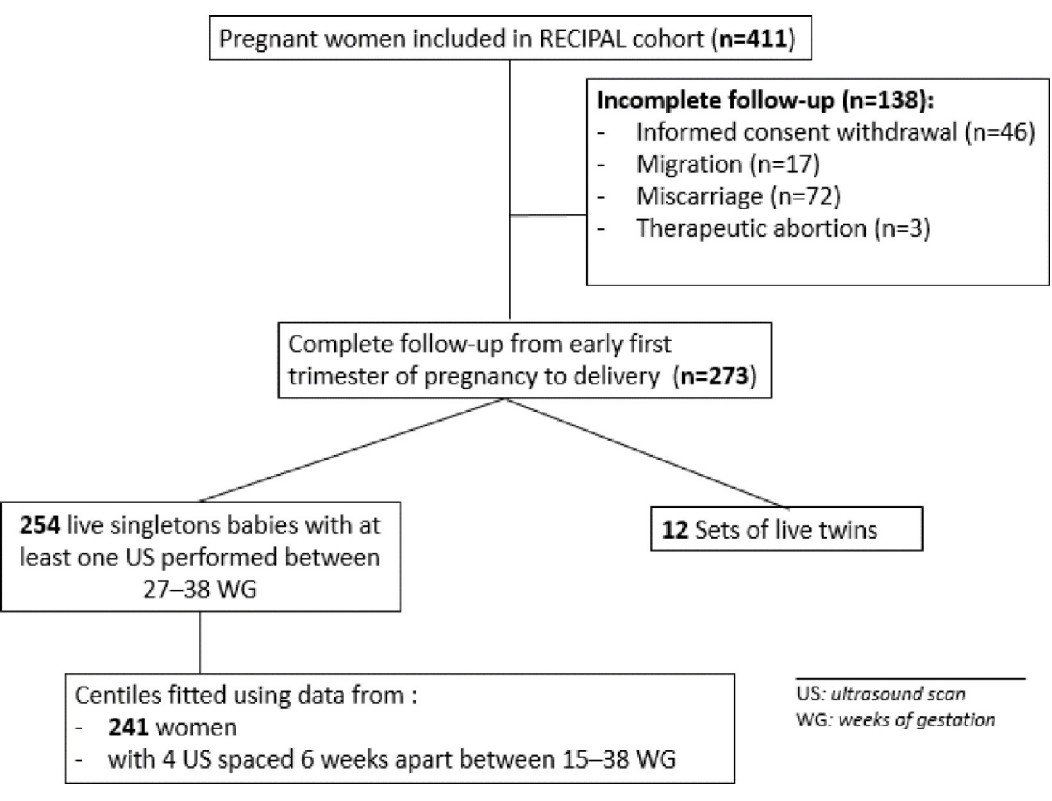

**Fig 1. Flowchart diagram of the study.** RECIPAL cohort, Southern Benin, 2014–2017.

Before becoming pregnant, 9% of women were underweight, and over half (57%) were anaemic (haemoglobin level <11g/L). Pregnancy was confirmed on average at 7.1wg, and women had an average 8.9 scheduled antenatal care (ANC) visits. During pregnancy, 43.1% and 69% of women had at least one episode of malaria infection and anaemia, respectively. Gestational hypertension (2.6%), smoking (<1%), and alcohol consumption (<1%) were infrequent. Multigravidae accounted for 88.8% of the study population. Women were considered overweight or obese in 23.9% of cases. The mean (SD) birthweight was 3031.13 (412.4) grams. There were 3 stillbirths (11.8 per 1000 live births) and 9.0% preterm births (Table 2).

## Comparison of RECIPAL fetal growth pattern to WHO, INTERGROWTH-21st, and Tanzania patterns

The graphic comparison of RECIPAL centiles of HC, AC and FL to those of INTER-GROWTH-21st and WHO charts is presented in Figs 2 and S2 and S3. Overall, the centiles for AC in the RECIPAL cohort were closer to WHO than INTERGROWTH-21st as shown in Fig 2 (equations and fitted values for RECIPAL cohort given in S1-S6 Tables in S1 File). For 15–35wg, RECIPAL AC centiles were higher than INTERGROWTH-21st centiles and globally lower than WHO centiles. For EFW, the deviation observed between our study centiles and those of WHO and INTERGROWTH-21st was globally larger than what was observed for AC (Figs 3–6). For EFW, RECIPAL centiles were closer to WHO centiles (Fig 3) than the two INTERGROWTH-21st centiles using IG21st and Hadlock's formula suitably (Figs 4 and 5). We confirmed that the use of Hadlock EFW formula with IG21st yielded a

**Table 1. Characteristics of the 254 pregnant women included in the ultrasound study.** RECIPAL cohort, Southern Benin, 2014–2017.

| Characteristics | Category | Mean (SD) or % |
|---|---|---|
| **Age (years)** | | 26.7 (4.9) |
| | < 24 y | 24.2% |
| | 24–30 y | 55.6% |
| | > 30 y | 20.2% |
| **Ethnic group** | Toffin | 74.3% |
| | Fon | 7.7% |
| | Aïzo | 12.9% |
| | Others [a] | 5.1% |
| **Education** | Illiterate | 71.7% |
| **Socioeconomic status** * | Low | 34.1% |
| | Mild | 40.6% |
| | High | 25.3% |
| **Gravidity** | 1 | 11.2% |
| | 2 | 16.2% |
| | 3 | 22.0% |
| | ≥4 | 50.6% |
| **Pre-pregnancy BMI (kg/m$^2$)** | | 22.8 (4.2) |
| | < 18.5 | 8.9% |
| | 18.5–25 | 67.2% |
| | ≥ 25 | 23.9% |
| **Anaemia before conception** | Yes | 57.2% |
| **Height (cm)** | | 158.3 (6.2) |
| **Short stature (height < 155 cm)** | Yes | 27.4% |
| **Malaria infection before conception** | Yes | 5.9% |
| **Number of ANC visits during pregnancy [b]** | | 8.9 (1.8) |
| **Gestational age at the first ANC visit (weeks) [c]** | | 7.1 (2.6) |
| **Gestational hypertension** | ≥ 1 episode(s) | 2.6% |
| **Anaemia during pregnancy** | ≥ 1 episode(s) | 69.5% |
| **Malaria infection during pregnancy** | ≥ 1 episode(s) | 43.1% |
| **Clinical malaria infection during pregnancy [d]** | ≥ 1 episode(s) | 22.1% |

Abbreviations: SD, standard deviation; IQR, interquartile range; ITN, insecticide-treated bed net; BMI, body mass index; ANC visit, antenatal care visit.

* Socioeconomic status was approximated using a synthetic score combining occupation and ownerships of assets, which was then categorized according to the tertiles in the whole RECIPAL cohort.

[a] Other ethnic groups: Yoruba, Adja, Goun, Ahoussa, Cotafon, Mahi, Sahoue.

[b] Including both scheduled and unscheduled visits.

[c] Gestational age was estimated using ultrasound scan or last menstrual period.

[d] Positive thick blood smear or positive rapid diagnostic test with an axillary temperature ≥ 37.5°C or history of fever in the last 24 hours.

much larger difference between RECIPAL and INTERGROWTH-21$^{st}$ centiles (Figs 4 and 5). The comparison between RECIPAL and Tanzanian EFW centiles showed that the 50$^{th}$ and 90$^{th}$ RECIPAL centiles were higher than in the Tanzanian chart (Fig 6). In contrast, the 10$^{th}$ centile was similar for both charts between 25 and 36 wg, after which a decrease was observed for the Tanzanian centiles.

**Table 2. Characteristics at birth of the 254 newborns included in the ultrasound study.** RECIPAL cohort, Southern Benin, 2014–2017.

| Characteristics | Category/Definition | Mean (SD) or n (%) |
|---|---|---|
| Gender | Male | 135 (53.1) |
| Stillbirth | Per 1000 live births | 3 (11.8 ‰) |
| Gestational age at birth (weeks) | | 39.0 (1.7) |
| Preterm birth * | <37 weeks' GA | 22 (8.7%) |
| Birthweight (g) | | 3031 (412) |
| Low birthweight‡ | < 2500 g | 23 (9.0%) |
| Birth length (cm) | | 48.2 (2.6) |
| Birth head circumference (cm) | | 34.0 (1.5) |

* Preterm birth defined as childbirth before 37 weeks of gestation.

‡ Low birthweight: birthweight < 2500 g, stillbirth and twins excluded.

## Comparison of RECIPAL centiles with WHO, INTERGROWTH-21ˢᵗ, and Tanzania centiles using the Bland and Altman percentage difference method

For the biometric measurements, the lowest biases were consistently observed when comparing RECIPAL to WHO centiles than when comparing RECIPAL to INTERGROWTH-21ˢᵗ centiles. Figs 7 and 8 display comparisons of the different charts using the Bland-Altman for the AC and EFW over the GA spectrum and provide the coefficients for the mean differences (S4 and S5 Figs show results for HC and FL). Regardless of the standard considered, the percentage differences were generally greater in early pregnancy with a gradual reduction until

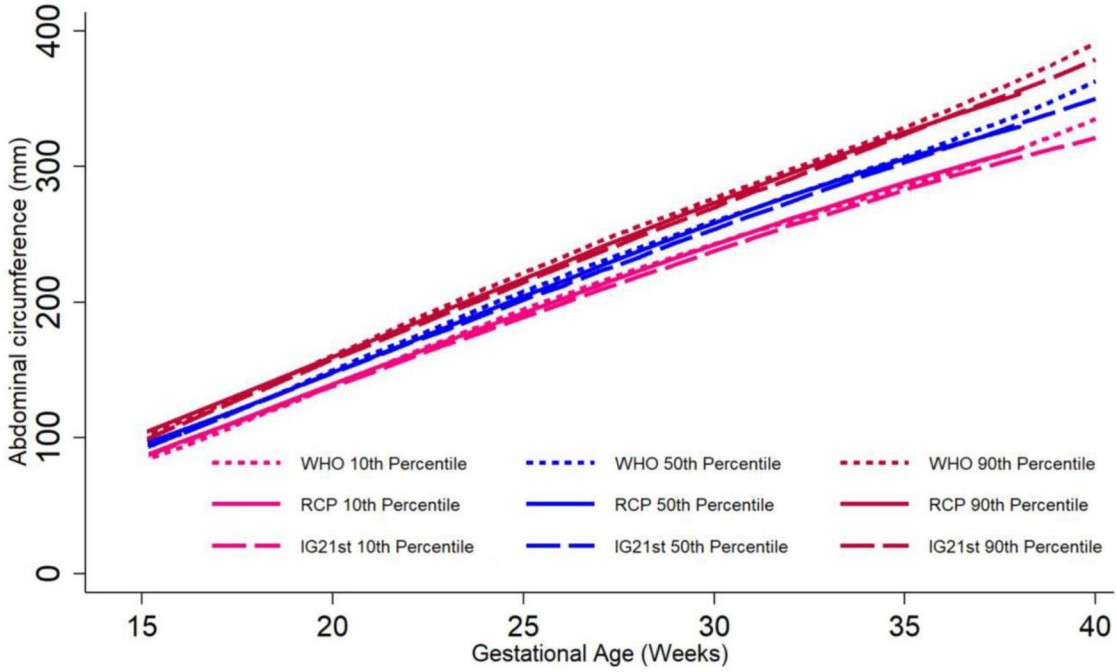

**Fig 2. Graphic comparison of RECIPAL study 10ᵗʰ, 50ᵗʰ, and 90ᵗʰ centiles of abdominal circumference (AC) to those of INTERGROWTH-21ˢᵗ and WHO charts.** RECIPAL cohort, Southern Benin, 2014–2017.

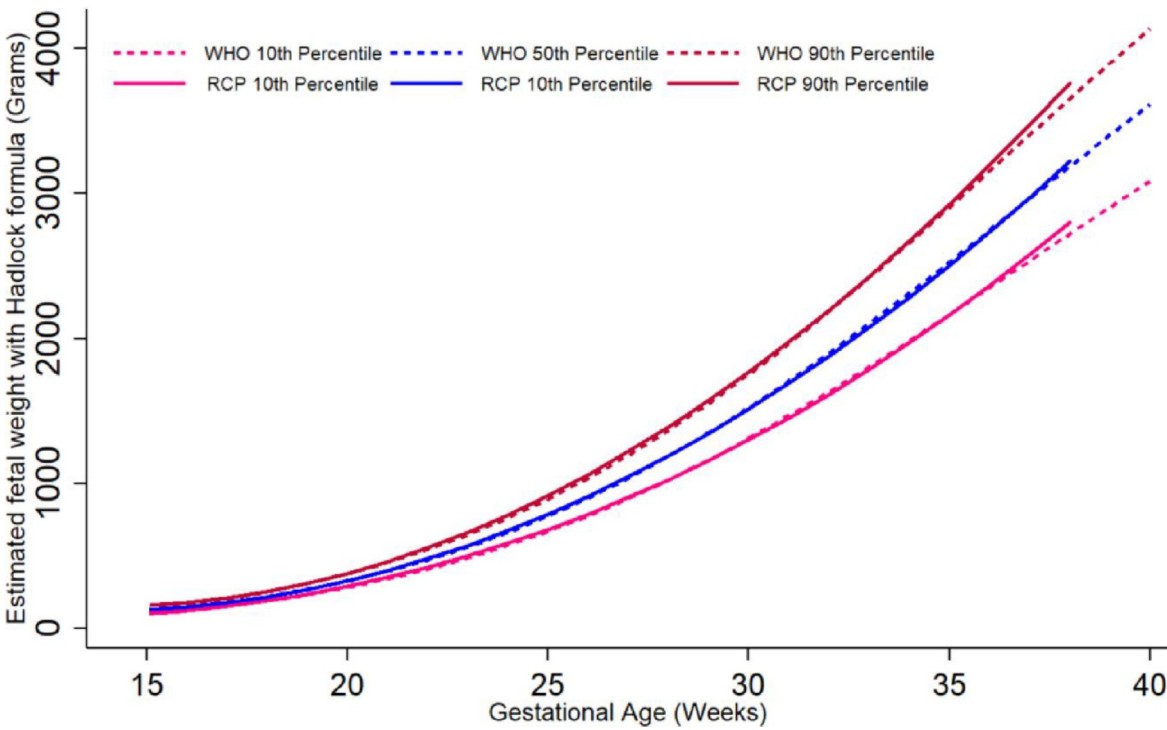

**Fig 3. Graphic comparison of RECIPAL study 10th, 50th, and 90th centiles of estimated fetal weight (EFW) to that of WHO charts.** Both RECIPAL study and WHO EFWs were calculated using the Hadlock formula. RECIPAL cohort, Southern Benin, 2014–2017.

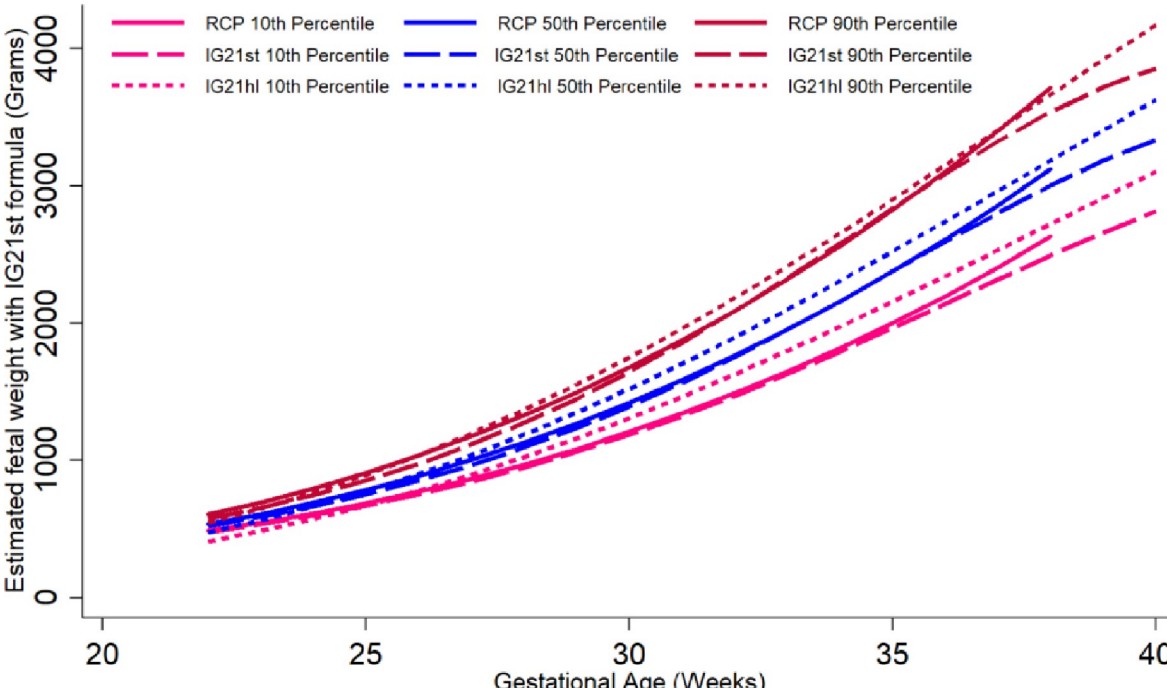

**Fig 4. Graphic comparison of RECIPAL study 10th, 50th, and 90th centiles of estimated fetal weight (EFW) to that of INTERGROWTH-21st charts.** RECIPAL study and INTERGROWTH-21st EFWs were calculated using the INTERGROWTH-21st formula. RECIPAL cohort, Southern Benin, 2014–2017.

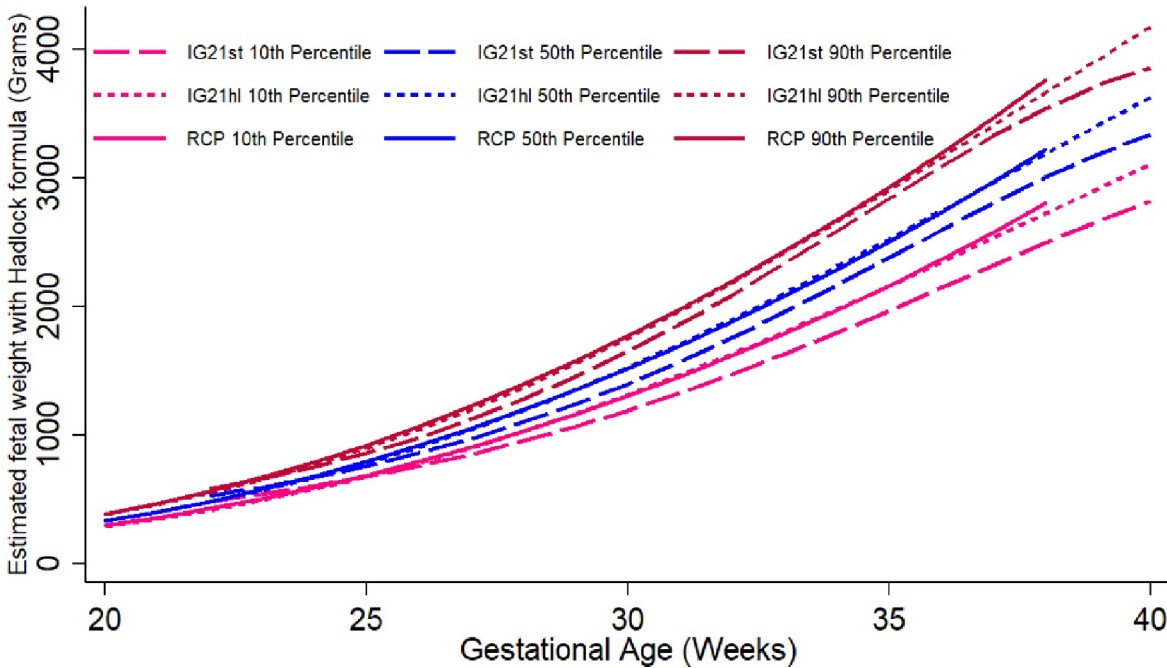

**Fig 5. Graphic comparison of RECIPAL study 10th, 50th, and 90th centiles of estimated fetal weight (EFW) to that of INTERGROWTH-21st recent charts using Hadlock formula.** RECIPAL study and INTERGROWTH-21st EFWs were calculated using the Hadlock formula as the recent IG21hl. RECIPAL cohort, Southern Benin, 2014–2017.

the end of the pregnancy. Regarding the 10th centile of EFW, RECIPAL and WHO values were quite similar (percentage difference of -0.23%), INTERGROWTH-21st (IG21st) values were on average 2.24% lower than those of RECIPAL, and -5.0% lower than RECIPAL for the recent IG21hl using Hadlock formula (Fig 8A). Tanzanian values were on average 2.62% lower than those of RECIPAL (Fig 8A). For the 50th and 90th centiles, the observed differences remained of the same magnitude for WHO but were greater for INTERGROWTH-21st and the Tanzanian chart (Fig 8B and 8C).

## Prevalence of biometric measurements and EFW <10th percentile according to the standard used

The proportion of fetuses with AC, FL, HC, or EFW <10th centile was higher when using WHO compared to INTERGROWTH-21st standards whenever assessed during pregnancy (Table 3). The new INTERGROWTH-21st centile values for EFW using the Hadlock formula performed similarly as with their own formula. As an internal validation of the modeled centiles using RECIPAL data, the proportion of fetuses with AC, FL, HC, or EFW <10th centile were close to the expected 10% (Table 3 and S7 Table in S1 File).

## Discussion

Using high-quality and prospectively collected ultrasound data early in gestation until delivery in a Beninese population, this study provides novel data assessing the two new international fetal growth standards in an African context. Contrary to expectations, RECIPAL cohort centiles were higher than INTERGROWTH-21st centiles and were close to those of WHO globally despite high rates of malarial infection and anaemia. RECIPAL 10th centile for EFW was

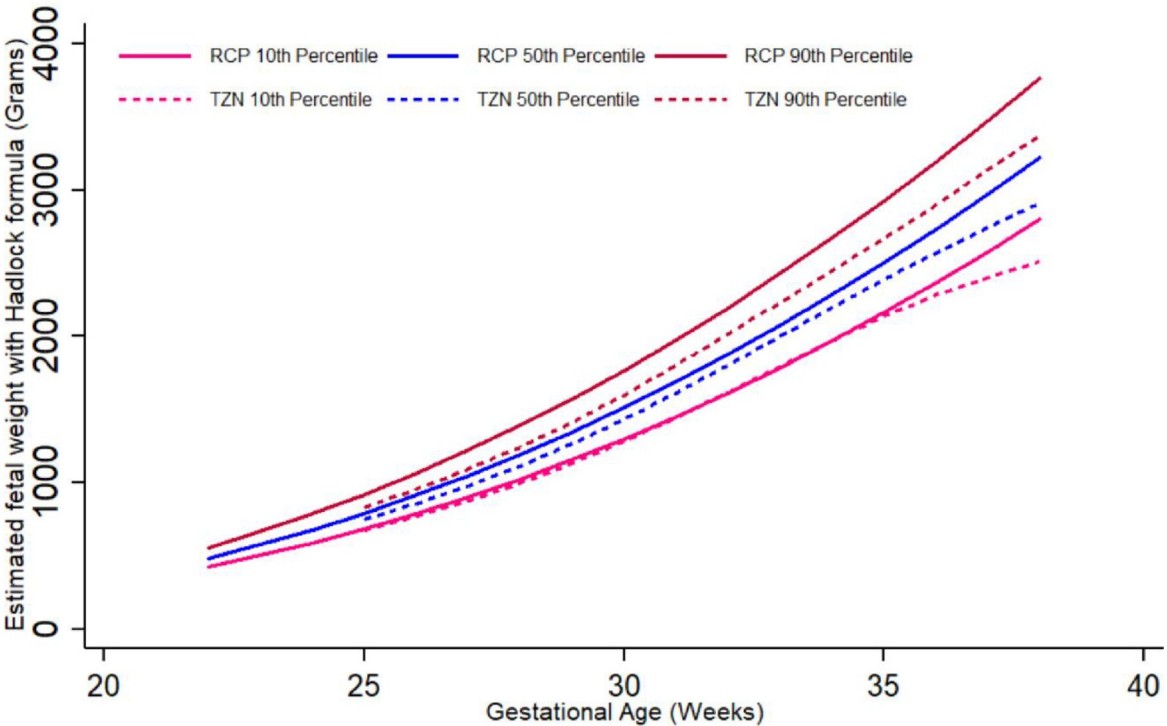

**Fig 6. Graphic comparison of RECIPAL study 10th, 50th, and 90th centiles of estimated fetal weight (EFW) to that of Tanzanian charts.** Both RECIPAL study and Tanzanian charts EFW were calculated using the Hadlock formula. RECIPAL cohort, Southern Benin, 2014–2017.

similar to that of Tanzania for 25–30wg, after which the difference increased. All these trends were confirmed by the percentage difference analysis and by assessing proportions of fetuses with measures below the 10th percentile.

While high-risk pregnancies were excluded from the WHO, INTERGROWTH-21st, and Tanzanian samples, they were strongly represented in RECIPAL sample with 9% underweight women, between 40% and 70% of women with anaemia or malaria during pregnancy, and an overall proportion of low birth weight of 9% [13, 14]. Therefore, the observed differences between RECIPAL, international, and Tanzanian centiles were smaller than expected. The high proportion of overweight/obese (24%) and multigravid (72%) women may partly explain the high RECIPAL centiles [32, 33]. Also, the optimal follow-up and management of women in our cohort may have contributed to reducing the impact of the various risk factors for FGR. In particular, women were screened for microscopic malaria every month and those infected were treated immediately. This may have mitigated the impact of malarial infection on fetal growth which would be an encouraging result for pregnant women living in high-malaria areas. We could not conduct a sensitivity analysis in women at lower risk of FGR because only 60 women (23%) met the stringent INTERGROWTH-21st and WHO criteria.

We applied both INTERGROWTH-21th and Hadlock formula to estimate fetal weight in our population and our comparison of the two INTERGROWTH-21th standards based on their own and Hadlock formula for EFW provides novel information about these two charts. The INTERGROWTH-21st formula has been found to underestimate fetal weight [18] and this was corroborated in our sample. Both INTERGROWTH-21th standards yielded centiles

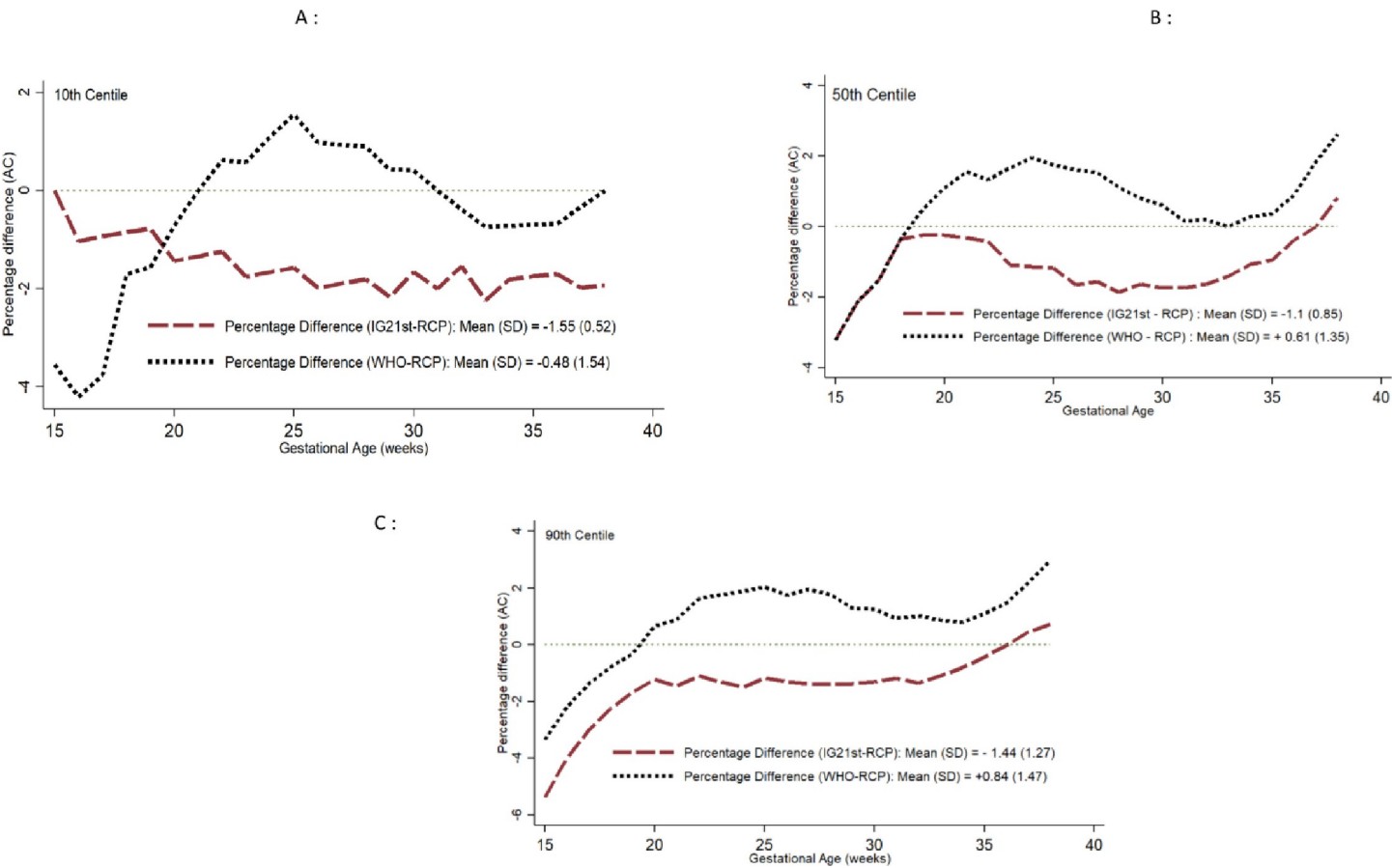

**Fig 7.** Percentage difference in 10th (A), 50th (B), and 90th (C) centiles of abdominal circumference (AC) between RECIPAL centiles and those of WHO and INTERGROWTH-21st. RECIPAL cohort, Southern Benin, 2014–2017.

that were lower than the RECIPAL centiles, with a larger gap for the standards using the Hadlock formula [15, 18, 34].

Limited African data were included in the development of INTERGROWTH-21st (data from Kenya) and WHO (Egypt and Democratic Republic of Congo) charts, although they are recommended for universal use. To our knowledge, only one study has evaluated these new standards in Africa. In Ethiopia, based on 675 singleton pregnancies, local fetal growth patterns (as illustrated by the 50th centile of AC, HC, FL and EFW) were reported to have the same distributions as those from the two international charts [22]. These findings from Ethiopia are difficult to interpret, given the differences between the charts. Furthermore, one might have expected a lower agreement between the two charts since the Hadlock formula was used in both cases to estimate fetal weight. In addition, while their study population included in a high proportion of undernurished women, the 5th local EFW centiles was higher than that of the WHO.

In HICs, studies have led to contradictory results. In Italy, Bellussi *et al.* concluded that INTERGROWTH-21st and local AC standards were interchangeable for the diagnosis of SGA fetuses [20]. In France, in their study including over 4,800 low-risk pregnancies, Stirnemann *et al.* concluded that French HC centiles closely matched INTERGROWTH-21st centiles. However, they did not provide a comparison of the centile references for AC and FL, for

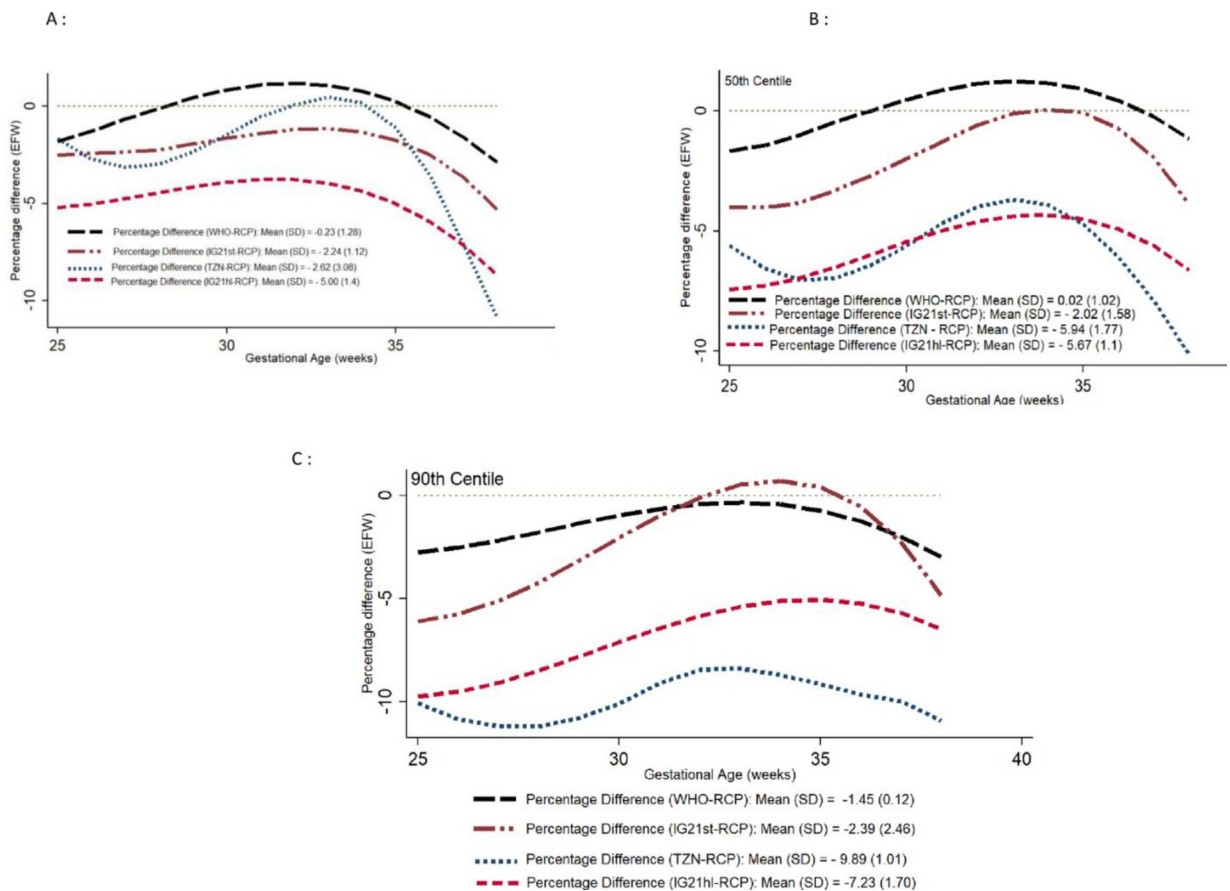

**Fig 8.** Percentage difference in 10<sup>th</sup> (A), 50<sup>th</sup> (B), and 90<sup>th</sup> (C) centiles of estimated fetal weight (EFW) between RECIPAL centiles and those of WHO, INTERGROWTH-21<sup>st</sup> (IG21st when using INTERGROWTH-21<sup>st</sup> EFW formula and IG21hl when using the Hadlock formula), and Tanzania. RECIPAL cohort, Southern Benin, 2014–2017.

which the proportions below the 10<sup>th</sup> centile were far lower than 10% [17]. AC and FL discrepancies between French and INTERGROWTH-21<sup>st</sup> centiles were also demonstrated using a very large sample of low-risk pregnancies belonging to the French Elfe cohort [35]. Similar results were published by Cheng *et al.*, who showed large differences between the INTERGROWTH-21<sup>st</sup> standards and the Chinese biometry standards using data collected on more than 10,000 unselected pregnancies. In particular, they found larger proportions of AC, HC, and FL less than 10<sup>th</sup> using INTERGROWTH-21<sup>st</sup> compared to their own standards, without a significant increase in the number of very SGA newborns at birth (14). Finally, in accordance with WHO that there may be significant differences between countries, recent studies in HICs also argued in favor of ethnic/geographic-specific fetal growth patterns [15, 21].

The RECIPAL study has several important strengths. In particular, women were recruited and followed up from the preconception period, allowing for accurate dating of the pregnancy by early US. Also, fetal growth was monitored prospectively throughout the pregnancy, making RECIPAL one of the few cohorts in Africa with such high-quality data. However, several limitations must also be considered, the main one being the small size of our study sample. Also, measurements of the fetuses were not equally distributed at each GA throughout the pregnancy, with four peaks at 16, 22, 24, and 28wg, and a low number of USs were performed after 35wg. Therefore, growth patterns starting from 35wg, and particularly the decrease in RECIPAL centiles for AC, must be interpreted with caution. Finally, because of our relatively

**Table 3. Proportion of fetuses below the 10th percentile according to INTERGROWTH-21st and WHO standards for HC, AC, FL and EFW at 27-32weeks and 33–38 weeks.**

| Biometric parameter | | Number (%) of fetuses with values < 10th percentile | | | | |
|---|---|---|---|---|---|---|
| | | RECIPAL | TANZANIA | IG 21hl | IG 21st | WHO |
| | | n (%) | n (%) | (based on Hadlock formula for EFW) | (based on INTERGROWTH formula for EFW) | n (%) |
| | | | | n (%) | n (%) | |
| HC (N = 243) | 27–32 weeks [a] | 21 (8.6) | - | - | 10 (4.1) | 18 (7.4) |
| HC (N = 232) | 33–38 weeks [b] | 23 (9.9) | - | - | 22 (9.5) | 22 (9.5) |
| AC (N = 243) | 27–32 weeks [a] | 26 (10.7) | - | - | 15 (6.2) | 33 (13.6) |
| AC (N = 232) | 33–38 weeks [b] | 23 (9.9) | - | - | 11 (4.7) | 16 (6.9) |
| FL (N = 243) | 27–32 weeks [a] | 27 (11.1) | - | - | 5 (2.1) | 10 (4.1) |
| FL (N = 232) | 33–38 weeks [b] | 20 (8.6) | - | - | 6 (2.6) | 17 (7.3) |
| EFW (N = 243) | 27–32 weeks [a] | 24 (9.9) | 16 (6.6) | 13 (5.4) | 18 (7.4) | 26 (10.7) |
| EFW (N = 232) | 33–38 weeks [b] | 24 (10.3) | 20 (8.6) | 10 (4.3) | 13 (5.6) | 27 (11.6) |

AC: Abdominal Circumference, HC: Head Circumference, EFW: Estimated Fetal Weight using 1) the Hadlock formula for comparison with WHO fetal growth standard (Kiserud et al., PLoS MED 2017) and with IG21hl standards (Stirnemann et al.; UOG 2020), and 2) the INTERGROWTH21-st formula for comparison with INTERGROWTH-21st fetal growth standard (Papageorghiou et al., Lancet 2014).

[a] Ultrasound scan performed within 27–32 weeks,

[b] Ultrasound scan performed within 33–38 weeks. IG21: INTERGROWTH-21st.

small sample size, we were unable to develop RECIPAL charts in a selected group of women at low risk. For these reasons, RECIPAL centiles were computed for descriptive and comparative purposes only and not as possible references for Benin.

In conclusion, WHO fetal growth charts seemed to better reflect the Beninese population than INTERGROWTH-21st charts, whatever the standards (IG21st vs. IG21hl) used. However, this finding needs to be confirmed on a larger sample, preferably restricted to low-risk pregnancies. Comparison of these international standards with high-quality African reference charts [11] is also warranted. Such future studies should evaluate to what extent these standards make it possible to identify children at risk of morbidity. In addition to samples of low risk pregnancies, investigations should consider application of these charts in sub-groups at risk, for instance, very low birthweight or preterm births. This is particularly important in African countries where the proportion of SGA newborns is estimated to be as high as 25% and the human and financial resources for the management of these children are limited. Finally, given the limited accuracy of fetal growth standards for identifying fetuses at risk of adverse perinatal outcomes [36], other strategies combining fetal biometry with biomarkers for FGR and women's clinical characteristics may find their place in the future [37]. Feasibility and cost-effectiveness will be important determinants of the large-scale use of this type of diagnostic tool in Africa.

## Supporting information

**S1 Fig. Distribution of ultrasound scans throughout pregnancy for fetal growth assessment.** RECIPAL cohort, Southern Benin, 2014–2017.
(TIF)

**S2 Fig. Graphic comparison of RECIPAL study 10th, 50th, and 90th centiles of head circumference (HC) to those of INTERGROWTH-21st and WHO charts.** RECIPAL cohort, Southern Benin, 2014–2017.
(TIF)

**S3 Fig. Graphic comparison of RECIPAL study 10[th], 50[th], and 90[th] centiles of femur length (FL) to those of INTERGROWTH-21[st] and WHO charts.** RECIPAL cohort, Southern Benin, 2014–2017.
(TIF)

**S4 Fig.** Percentage difference in 10th (A), 50th (B), and 90th (C) centiles of head circumference (HC) between RECIPAL centiles and those of WHO and INTERGROWTH-21st. RECIPAL cohort, Southern Benin, 2014–2017.
(TIF)

**S5 Fig.** Percentage difference in 10th (A), 50th (B), and 90th (C) centiles of femur length (FL) between RECIPAL centiles and those of WHO and INTERGROWTH-21st. RECIPAL cohort, Southern Benin, 2014–2017.
(TIF)

**S1 File. Supporting tables. Supplementary Table S1:** Equations for the estimation of each percentile using quantile regression of each fetal measurement (in mm) according to gestational age (in weeks). **Supplementary Table S2:** Centiles of abdominal circumference (N = 241). RECIPAL cohort, Southern Benin, 2014–2017. **Supplementary Table S3:** Centiles of head circumference (N = 241). RECIPAL cohort, Southern Benin, 2014–2017. **Supplementary Table S4:** Centiles of estimated fetal weight using the Hadlock formula (N = 241). RECIPAL cohort, Southern Benin, 2014–2017. **Supplementary Table S5:** Centiles of estimated fetal weight using the INTERGROWTH-21st formula starting from 22 weeks of gestation (N = 241). RECIPAL cohort, Southern Benin, 2014–2017. **Supplementary Table S6:** Centiles of femur length (N = 241). RECIPAL cohort, Southern Benin, 2014–2017. **Supplementary Table S7:** The proportion of fetuses with observed values below the threshold of each percentile (Q) using quantile regression to model RECIPAL data, RECIPAL cohort, Southern Benin, 2014–2017.
(DOCX)

## Acknowledgments

The authors thank all the local communities of Sô-Ava and Akassato in Benin who took part in this study; all RECIPAL project technical team, the midwives, nurses, community-health workers for the hard work of recruiting and following the study participants.

## Author Contributions

**Conceptualization:** Emmanuel Yovo, Alice Hocquette, Barbara Heude, Jennifer Zeitlin, Valérie Briand.

**Data curation:** Emmanuel Yovo, Manfred Accrombessi, Gino Agbota, Nadine Fievet.

**Formal analysis:** Emmanuel Yovo, Alice Hocquette.

**Funding acquisition:** Valérie Briand.

**Investigation:** Emmanuel Yovo, Manfred Accrombessi, Gino Agbota, William Atade, Olaiitan T. Ladikpo, Murielle Mehoba, Auguste Degbe, Valérie Briand.

**Methodology:** Emmanuel Yovo, William Atade, Ghyslain Mombo-Ngoma, Achille Massougbodji, Nikki Jackson, Jennifer Zeitlin, Valérie Briand.

**Project administration:** Manfred Accrombessi, Valérie Briand.

**Resources:** Valérie Briand.

**Software:** Emmanuel Yovo.

**Supervision:** Emmanuel Yovo, Manfred Accrombessi, Valérie Briand.

**Validation:** Emmanuel Yovo, Nikki Jackson, Jennifer Zeitlin, Valérie Briand.

**Visualization:** Emmanuel Yovo.

**Writing – original draft:** Emmanuel Yovo.

**Writing – review & editing:** Manfred Accrombessi, Gino Agbota, Alice Hocquette, William Atade, Olaiitan T. Ladikpo, Murielle Mehoba, Auguste Degbe, Ghyslain Mombo-Ngoma, Achille Massougbodji, Nikki Jackson, Nadine Fievet, Barbara Heude, Jennifer Zeitlin, Valérie Briand.

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
