## [Decision Letter · Decision Letter 0]

13 May 2021

PONE-D-21-11431

Assessing fetal growth in Africa: application of the international WHO and INTERGROWTH-21st standards in a Beninese pregnancy cohort

PLOS ONE

Dear Dr. YOVO,

Thank you for submitting your manuscript to PLOS ONE. After careful consideration, we feel that it has merit but does not fully meet PLOS ONE’s publication criteria as it currently stands. Therefore, we invite you to submit a revised version of the manuscript that addresses the points raised during the review process.

This manuscript has merit but I agree with the reviewers that revisions need to be made to improve it. Whilst all the points raised by the reviewers need to be either dealt with or robustly rebutted, I think that the most important of these are the first point from reviewer 1 (about the need to transform the data or use non-parametric analysis techniques such as quantile regression) and the point about separating results from healthy pregnancies from those affected by e.g. anaemia/malaria, made by reviewers 2 and 3.

In addition to the points raised by the reviewers, and although the authors state that the data from the study is fully available (one of the publication criteria of this journal), I can not find a file containing the raw data or a link to where I can download it. This does not fit with the journal publication criteria. The data either needs to be included within the revision or a link to a public repository that contains the data needs to be provided.

We look forward to receiving your revised manuscript.

Kind regards,

Clive J Petry, PhD

Academic Editor

PLOS ONE

Journal Requirements:

2) PLOS requires an ORCID iD for the corresponding author in Editorial Manager on papers submitted after December 6th, 2016. Please ensure that you have an ORCID iD and that it is validated in Editorial Manager. To do this, go to ‘Update my Information’ (in the upper left-hand corner of the main menu), and click on the Fetch/Validate link next to the ORCID field. This will take you to the ORCID site and allow you to create a new iD or authenticate a pre-existing iD in Editorial Manager. Please see the following video for instructions on linking an ORCID iD to your Editorial Manager account: https://www.youtube.com/watch?v=_xcclfuvtxQ

Reviewers' comments:

Reviewer's Responses to Questions

**Comments to the Author**

1. Is the manuscript technically sound, and do the data support the conclusions?

Reviewer #1: Partly

Reviewer #2: Yes

Reviewer #3: No

2. Has the statistical analysis been performed appropriately and rigorously? 

Reviewer #1: No

Reviewer #2: Yes

Reviewer #3: No

3. Have the authors made all data underlying the findings in their manuscript fully available?

Reviewer #1: No

Reviewer #2: Yes

Reviewer #3: Yes

4. Is the manuscript presented in an intelligible fashion and written in standard English?

Reviewer #1: Yes

Reviewer #2: Yes

Reviewer #3: Yes

5. Review Comments to the Author

Reviewer #1: The authors assessed fetal biometric measurements collected longitudinally in 241 singleton pregnancies and compare the distribution of the data against two existing standards. The study could inform on the selection of the most appropriate standard for their population.

Major:

1)The approach used by the authors to determine the biometrics and EFW centiles 1) assumes data is normally distributed for given gestational age and 2) treats the longitudinal observations from the same subject as independent observations, which are not. Neither IG21, WHO nor other more recent studies (https://www.ncbi.nlm.nih.gov/pmc/articles/PMC5815382/) rely on such assumptions. Data should be transformed (e.g. Box-Cox transform or log for EFW) to enable normality assumption, or rely on quantile regression instead. Within subject correlations should be dealt with using appropriate techniques (e.g. mixed effects models) or other. At a minimum, these should be stated as limitations and figures based on such models should be moved to the supplement and rely on Table 3 cross-sectional results instead.

2) The analysis in Table 3 should be the main driver or results presented and discussion because they are free of the methodological imitations described in point 1 above. Based on table 3 cross-sectional analyses, EFW values observed herein tend to be higher than the cut-offs derived in the WHO and especially IG21 studies. SGA (<10th) at 33-38 weeks are 8.7% and 4.7% based on WHO and IG21, respectively compared to 10% (expected). Therefore, in this light, and considering comments at point 1, the abstract statement “The difference in both 10 th -centile AC and EFW between WHO and RECIPAL was positive starting from 20wg” is confusing. If EFW 10th centile would be higher in WHO than in this study after 20 weeks, I would expect at 33-38weeks, the rate of SGA <10th should be more than 10% for WHO, but it is only 8.7%. This inconsistency comes from issues outlined in 1.

Reviewer #2: This is an important study made on a optimal set of patients where information could be gathered from pre-conception until delivery.

The serial US is highly informative since it can precise dating - since the earliest US is the most accurate for dating.

Of note that this cohort got optimal care with monitoring monthly malaria infection and treating promptly the patient as soon it is identified. Clearly otherwise outcome would be different.

As such it does not fully represent the high risk population where diagnosis and treatment are limited.

What was important to note that prematurity and stillbirth rates were low- which indicates that adequate management can have significant beneficial effects.

In general access to US is limited especially in time of Covid and also in LMIC therefore a specific time point when US is most accurate to predict IUGR should be better described is it at 28 weeks or after 33 weeks where the margin of error does significantly increase.

For maximizing resources do authors suggest based on their findings how to allocate those resources. For example if urine pregnancy test is instituted and carefully implemented which US if only one time point available should be used.

Overall, it is a well documented study where comparison with other charts of fetal growth are assessed.

Although this population is defined as low risk - it is not. Therefore comparison with chart of healthy patients should be viewed in caution.

Separating the chart between those with malaria/anemia and other without could provide a better insight since it could be compared to low risk population in other charts.

But it is the local reality in that region which also applies to several other African countries so it has to be accepted reality - the best analysis that can be carried out under the circumstances.

The outcome of these pregnancies - since access to newborn care may be limited could also define at what point such US observation when newborn size is <2500 the risk is at high for morbidity. This is clearly different from high resource countries where prematurely born can survive and and develop even when <25 w gestation.

This is especially important if induction is contemplated due to IUGR. Comment on this needed.

Reviewer #3: This study is a potentially important examination of the applicability of 3 standards – 2 international (Intergrowth 21st and WHO) and one from Tanzania, on a maternity cohort in Benin. The local data were from antenatal ultrasound measurements collected in a previous study of a pre-conception cohort of pregnancies (Recipal).

The authors modelled growth from their Recipal database using a polynomial, and compared this with the three standards using percentile differences. The problem with this approach is that their cohort is a reference of the whole population including pathology, whereas the others are standards, based on normal populations excluding pathology. Their method makes it difficult to address the key objective put forward in the title, of how well each of the international standards perform in Benin, by 1. confirming growth in normal pregnancy and 2. identifying pathology associated with fetal growth restriction, such as malaria and anaemia.

The authors state that they did not have sufficient cases to fulfil the stringent criteria for normality used in WHO and IG21 (lines 277-8). However they do not need to equal these criteria to divide their cohort into normal and abnormal outcome pregnancies: this will then allow them to derive antenatal and neonatal SGA rates associated with normal and affected pregnancies according to the 3 standards being investigated, and comment on their suitability for their population. Currently, the method of analysis leaves too many unanswerable questions as to why differences exist.

Altman Bland analyses normally include correlation coefficients with 95% confidence intervals, and this should be included. It would also be interesting to see the systematic and random error for their EFW measurements: if there was a high degree of over-estimation, this could explain low SGA rates. It would also be good to see percent difference within gestational age categories – one would typically expect similar differences across the gestational ages, which was not the case here.

6. PLOS authors have the option to publish the peer review history of their article (what does this mean?). If published, this will include your full peer review and any attached files.

Reviewer #1: No

Reviewer #2: **Yes: **Eytan R Barnea MD FACOG

Reviewer #3: No

---

## [Author Response · Author response to Decision Letter 0]

17 Nov 2021

Response to Reviewer # 1

As suggested by the reviewer we reanalysed the RECIPAL data using quantile regression which does not make any assumption about the normality of the distribution. 

As was done by the WHO for developing their standards, correlation between measurements in a single woman was not taken into account. We consulted with the WHO statistical team for their reasoning, which was that mixed-effects models would not affect the estimations of the coefficients, while increasing complexity. 

Reassuringly, using this method resulted in only very small changes in the RECIPAL percentile values.

Thank you for these suggestions. We have redone this table and given it more prominence in the manuscript. 

Changes also include adding the proportion of fetuses with AC, HC, FL and EFW centiles <10th after RECIPAL modelling, which provides a check of internal validity for the modelling (above). 

We also added the recently published INTERGROWTH-21st EFW standards based on the Hadlock formula for estimating fetal weight. 

Response to Reviewer # 2

We thank the reviewer for the positive comments about our study. 

The reviewer is correct to state that women in the RECIPAL cohort had a better follow-up than Beninese women have in general. These women were tested repeatedly for malaria during their pregnancy and treated. This may be one explanation for our results finding a good correspondence with WHO and fewer fetuses under the 10th percentile than INTERGROWTH-21st . We did not expect this finding since the samples used to construct the charts by WHO and INTERGROWTH-21st were selected to be low-risk. We have added some further discussion of this point on page 16.

Unfortunately, our sample size was not large enough to carry-out meaningful sub-group analysis by risk status, either by defining the sub-population of low-risk women or only women who were malaria-free. 

We agree with the reviewer that at the end the standards (whichever they are) must be sensitive and specific enough to identify those foetuses who will be at higher risk of morbidity and mortality. Unfortunately, because of the low number of SGA foetuses we were not able to assess the clinical predictive value of each standard. We have added this as an important area for further research, especially in populations where the outcomes of the highest risk pregnancies could be analysed: on page 19.

Based on the new WHO guidelines, one US before 24 weeks of gestation is recommended. In addition, evolving evidence suggests that combining early US with a “late” third trimester US is beneficial for detecting/confirming third trimester-complications such as FGR and for improving facility-based delivery (Sovio et al., Lancet 2015). The evidence on the impact of US on maternal and perinatal mortality is contrasted to date (Goldenberg et al., BJOG 2018; Saari-Kemppainen et al., Lancet 1990), partly explained by sub-optimal study designs and missed opportunities for adequate clinical management of complications detected by US. Also, one may not exclude that standards used to identify growth-restricted foetuses were not optimal. As mentioned in the manuscript (page 19), “Future studies should evaluate to what extent these standards make it possible to identify children at risk of morbidity.” 

Response to Reviewer # 3

In response to the reviewer’s comments, we have clarified the principal purpose of our study, which was to assess the prevalence of FGR during pregnancy using international charts in our cohort. We recognize that our study population does not conform to the low-risk population used for the construction of the WHO and IG prescriptive charts and, indeed, our expectation was that we would find higher proportions of foetuses under the 10th percentile. 

This was not the case, which is the paper’s main message. Note that our aim was not to validate these charts against a low-risk population. Malaria, anaemia and maternal underweight are important risk factors for FGR and these are endemic in our population. As explained in our response to reviewer 2, we did not have a sufficient sample size to carry out sub-group analyses which were not planned. For instance, the prevalence of malaria (43%) and anaemia (69%) are high. If we add maternal underweight and clinical malaria, we reduce the sample of women eligible for analysis.

Based on the comments of reviewer 1, we have redone the calculation of the RECIPAL percentiles finding very similar results. We do not believe that there is over-estimation of the ultrasound measures since the study followed a strict protocol with many checks and we have no reason to believe that there would be systematic error in one direction.

---

## [Decision Letter · Decision Letter 1]

21 Dec 2021

PONE-D-21-11431R1Assessing fetal growth in Africa: application of the international WHO and INTERGROWTH-21st standards in a Beninese pregnancy cohortPLOS ONE

Dear Dr. YOVO,

Thank you for submitting your manuscript to PLOS ONE. After careful consideration, we feel that it has merit but does not fully meet PLOS ONE’s publication criteria as it currently stands. Therefore, we invite you to submit a revised version of the manuscript that addresses the points raised during the review process.

Thank you for submitting the revised version of the manuscript. We have managed to get two of the original reviewers to have another look at it. I agree with their opinions, that the manuscript still requires a little more revision to get it as good as possible. I would therefore like you to complete the revisions suggested by the two reviewers - I won't add anything further to them. I look forward to seeing what should be the final version of the manuscript, which I would enocurage you to submit at your earliest convenience.

We look forward to receiving your revised manuscript.

Kind regards,

Clive J Petry, PhD

Academic Editor

PLOS ONE

Reviewers' comments:

Reviewer's Responses to Questions

**Comments to the Author**

1. If the authors have adequately addressed your comments raised in a previous round of review and you feel that this manuscript is now acceptable for publication, you may indicate that here to bypass the “Comments to the Author” section, enter your conflict of interest statement in the “Confidential to Editor” section, and submit your "Accept" recommendation.

Reviewer #2: All comments have been addressed

Reviewer #3: All comments have been addressed

2. Is the manuscript technically sound, and do the data support the conclusions?

Reviewer #2: Partly

Reviewer #3: Yes

3. Has the statistical analysis been performed appropriately and rigorously? 

Reviewer #2: Yes

Reviewer #3: Yes

4. Have the authors made all data underlying the findings in their manuscript fully available?

Reviewer #2: Yes

Reviewer #3: Yes

5. Is the manuscript presented in an intelligible fashion and written in standard English?

Reviewer #2: Yes

Reviewer #3: Yes

6. Review Comments to the Author

Reviewer #2: The comments were addressed but Table 2 needs to be modified. Confusing.

it is written male,

The text is unclear birth weight = should be newborn weight ....

2500gm ?

Reviewer #3: 1. Authors' revision has addressed main point.

2. Coefficient for Altman Bland plot ought to be added.

3. Inclusion of new version of Intergrowth formula (using Hadlock for EFW) is a good step as it makes results 'current' with formula changes. However note that relevant reference for this correction by IG21 is 29, not 28.

4. Lines 256-7: IG21hl 'did not do better' is wrong term as there are no outcomes to use as a performance standard. Perhaps 'performed similarly' or 'had similar discrepancy to WHO' (as IG21)

5. Lines 331-2 As in 4, as no standard used/available, 'better match' needs to be reworded; perhaps 'seemed to better reflect...'

7. PLOS authors have the option to publish the peer review history of their article (what does this mean?). If published, this will include your full peer review and any attached files.

Reviewer #2: **Yes: **Eytan R Barnea MD

Reviewer #3: No

---

## [Author Response · Author response to Decision Letter 1]

2 Jan 2022

Response to Reviewer # 2

We thank the reviewer for the comment regarding the table 2. We have reorganised the table 2 with more details that make it easier to understand

We have referred to birth weight as the weight measured within minutes of the birth of each child.

Response to Reviewer # 3

 1. Authors' revision has addressed main point:

We thank the reviewer for the positive comments about the revised version of the paper. 

2. Coefficient for Altman Bland plot ought to be added.

Coefficients are shown in Figures 7 and 8, and Supplementary Figures 4 and 5

3. Inclusion of new version of Intergrowth formula (using Hadlock for EFW) is a good step as it makes results 'current' with formula changes. However, note that relevant reference for this correction by IG21 is 29, not 28.

Done,

Thank you very much for your attention which made it possible to correct the numbering error

4. Lines 256-7: IG21hl 'did not do better' is wrong term as there are no outcomes to use as a performance standard. Perhaps 'performed similarly' or 'had similar discrepancy to WHO' (as IG21)

Done,

Thank you for the rewording suggestions which are taken into account

5. Lines 331-2 As in 4, as no standard used/available, 'better match' needs to be reworded; perhaps 'seemed to better reflect...'

Done,

Thank you for the rewording suggestions which are taken into account

---

## [Editor Report · Decision Letter 2]

5 Jan 2022

Assessing fetal growth in Africa: application of the international WHO and INTERGROWTH-21st standards in a Beninese pregnancy cohort

PONE-D-21-11431R2

Dear Dr. YOVO,

We’re pleased to inform you that your manuscript has been judged scientifically suitable for publication and will be formally accepted for publication once it meets all outstanding technical requirements.

Kind regards,

Clive J Petry, PhD

Academic Editor

PLOS ONE
---

## [Editor Report · Acceptance letter]

11 Jan 2022

PONE-D-21-11431R2 

Assessing fetal growth in Africa: application of the international WHO and INTERGROWTH-21<sup>st<sup> standards in a Beninese pregnancy cohort 

Dear Dr. YOVO:

I'm pleased to inform you that your manuscript has been deemed suitable for publication in PLOS ONE. Congratulations! Your manuscript is now with our production department. 

Kind regards, 

on behalf of

Dr. Clive J Petry 

Academic Editor

PLOS ONE